# Muscle Metabolism During Multiple Muscle Stimulation Using an Affordable Equipment

**DOI:** 10.3390/jfmk9040248

**Published:** 2024-11-26

**Authors:** Samantha Ye, Sydney Stetter, Kevin K. McCully

**Affiliations:** Department of Kinesiology, University of Georgia, Athens, GA 30602, USA; samantha.ye@uga.edu (S.Y.); sydney.stetter@uga.edu (S.S.)

**Keywords:** near-infrared spectroscopy (NIRS), oxidative metabolism, skeletal muscle, neuromuscular electrical stimulation, human subjects

## Abstract

**Background/Objectives**: Previous studies have shown that neuromuscular electrical stimulation (NMES), while expensive, can provide some of the health benefits of exercise to people who cannot exercise their legs normally. The aim of this study was to quantify the increases in muscle metabolism in four muscles of the legs of able-bodied individuals with NMES. **Methods**: Healthy college-aged students were tested. NMES of four muscle groups was performed with inexpensive stimulators and reusable tin foil electrodes. The biceps femoris, vastus lateralis, medial gastrocnemius, and tibialis anterior muscles on one leg were stimulated for ten minutes with twitch stimulations at the highest comfortable stimulation current. Muscle metabolism was measured using the slope of oxygen consumption measured with near-infrared spectroscopy (NIRS) during 5 s of cuff ischemia. **Results**: Initial studies found fold increases in muscle metabolism above rest of 8.9 ± 8.6 for the vastus lateralis, 7.9 ± 11.9 for the biceps femoris, 6.6 ± 7.8 for the medial gastrocnemius, and 4.9 ± 3.9 for the tibialis anterior. Some participants were able to obtain large increases in muscle metabolism, while other participants had lower increases. **Conclusions**: The ability to produce large increases in metabolism has the potential to allow NMES to replace or augment exercise to improve health in people who cannot otherwise exercise. The devices used were inexpensive and could be adapted for easy use by a wide range of individuals.

## 1. Introduction

Physical fitness is necessary for the maintenance and attainment of optimal health. Exercise has been shown to reduce cardiovascular risk factors, including dyslipidemia, high blood pressure, and glucose tolerance [1,2]. Physical exercise is a major determinant of cardiovascular and metabolic risk, along with long-term disease mortality [3]. Guidelines to improve health include recommendations for physical activity levels [4]. In addition, recent research has added to these guidelines by suggesting that long, sedentary periods should be broken up with small bouts of physical activity [5].

Many people have conditions which decrease mobility or reduce the ability to be physically active [6,7,8]. Neuromuscular electrical stimulation (NMES), the application of electrical currents to motor nerves, has been used to contract skeletal muscle [9] and could potentially serve as a form of physical activity [10,11]. NMES of the muscles in the lower leg have provided exercise-related health benefits in patient populations, such as those with spinal cord injuries, chronic illnesses, or severe muscle atrophy [12,13,14,15]. NMES devices used to provide health benefits can stimulate six or more different muscle groups [12]. The stimulation patterns are often intricate and used to drive a cycle ergometer or to allow for a complex task to be performed, such as getting up off of a bed or out of a chair [12]. Other devices have been designed to work by adding supplemental muscle activation to normal exercise movements [16]. The complexity of the required stimulation patterns increases the cost of the devices. The stimulation devices and electrodes can be expensive enough to be impractical for widespread use. Given the high cost of traditional NMES devices, there is a critical need for affordable alternatives that maintain efficacy while expanding accessibility in both research and clinical contexts.

Traditionally, the response to exercise has been measured with whole-body indicators, such as changes in heart rate and whole-body oxygen consumption. Near-infrared spectroscopy (NIRS) has been developed as a noninvasive technique that measures changes in oxygenation and blood volume within skeletal muscle [17]. Using an acute arterial occlusion, NIRS can be employed to provide real-time data on muscle metabolism and oxygen utilization [18,19,20,21,22,23]. The NIRS approach lends itself well to evaluating changes in muscle metabolic rates with electrical stimulation.

The aim of this study was to increase muscle metabolism (mVO_2_) during multiple muscle stimulation using simple stimulation protocols and affordable electrical stimulation devices. The setup includes the use of aluminum foil electrodes instead of self-adhesive electrodes and a low-cost four-channel muscle stimulator. It was hypothesized that the electrical stimulation would produce an 8-fold increase in muscle metabolism over resting values. This elevation in muscle metabolism has been reported in previous studies of single muscle stimulation [24] and has been deemed to be a moderate exercise intensity.

## 2. Materials and Methods

### 2.1. Participants

The study used twenty healthy male and female subjects between 18–40 years of age. Subjects were chosen to represent a wide variety of muscle oxidative capacity and included college-aged endurance-trained and sedentary individuals. Participants were classified as ‘healthy’ if they reported no history of chronic illnesses, cardiovascular or metabolic diseases, musculoskeletal disorders or any condition that could impair physical function or metabolism at the time of the study. Healthy participants were selected to validate the feasibility and effectiveness of the protocol under controlled conditions, minimizing variability introduced by underlying muscle pathology. All studies were conducted with approval by the Institutional Review Board at the University of Georgia (Athens, GA, USA), and all subjects were given written, informed consent and asked to sign a consent form before testing. Subjects were recruited from the local community via word of mouth and email.

### 2.2. Experimental Design

A one group, one-day, pre-post experimental design with multiple measurements was conducted. The stimulation protocol consisted of 10 min of electrical stimulation of four muscle groups on one leg. Measurements of muscle metabolism were made in four muscle groups before and 5 min after the electrical stimulation. Measurements were also made at 5 and 10 min of electrical stimulation.

Each subject was placed on a padded table, supine, with both legs positioned straight (Figure 1). The left foot was secured in a stabilization holder on the padded table to limit motion artifacts in the NIRS signal. The knee was supported by a cushion. Measurements to find muscle oxygen levels were taken prior to and immediately following the electrical stimulation protocol. The experimental setup for the NIRS measurements used a continuous wave NIRS device (Train.Red Plus, Eisteinweg, The Netherlands). The NIRS data were collected at 10 Hz. The four NIRS probes were placed on the subject’s left biceps femoris (BF), vastus lateralis (VL), medial gastrocnemius (MG), and tibialis anterior (TA) muscles.

mVO_2_ was measured using short periods of ischemia [24]. A blood pressure cuff (Hokanson SC12D cuff inflator, Bellevue, WA, USA) was placed on the proximal thigh, as high as anatomically possible, to induce transient ischemia for the measurements of muscle metabolism. The cuff was fit to the circumference of the limb. The vascular cuff was attached to a Rapid Cuff Inflation system (Hokanson A10 and E20, Bellevue, WA, USA). For the pre-stimulation rest and post-stimulation recovery measurements, 30 s of ischemia were used. For the 5 and 10 min time points during electrical stimulation, 5 s of ischemia were used. Ischemia was produced by rapid cuff inflation at 260 mmHg. Metabolic rate was the slope of the decrease in the SmO_2_ value from the NIRS device.

Stimulation electrodes made of aluminum foil [25] were placed proximally and distally on the biceps femoris, vastus lateralis, medial gastrocnemius, and tibialis anterior. The electrodes were customized in length, and longer electrodes were used on the lower leg muscles while the shorter electrodes were used on the upper muscles. Ultrasound gel was used as a conduction medium between all electrodes and the skin. Pre-wrap was used to keep electrodes in the proper position. The stimulation frequency was set to approximately 3 Hz based on a preprogrammed setting from the electrical stimulation device (HealthmateForever Model#YK15AB, Lenexa, KS, USA). The current levels were adjusted to produce a visible, vigorous twitch contraction at the maximum tolerable level for each participant and ranged from levels 2 to 4 out of a scale of 10. The highest tolerable level of stimulation was encouraged to ensure muscle activation at the NIRS measurement site. The stimulation device used offered four channels that could be adjusted to two different intensities. Though the intensities were frequently uniform, the vastus lateralis and biceps femoris were set to one intensity, and the medial gastrocnemius and tibialis anterior received the same intensity. The stimulation mode used was the tapping setting. A speed of 25 clicks, which converts to about 3 Hz, was applied. The device’s 3 Hz stimulation protocol included occasional brief pauses in stimulation.

Prior to starting the protocol, a stimulation check was performed on all muscles to confirm appropriate muscle activation. After 30 s of NMES at 3 Hz, a blood pressure cuff was inflated to 260 mmHg for 30 s. This value represents zero oxygenation in the tissue under the NIRS probe. Upon release of the blood pressure cuff, the hyperemic response causes an overshoot in the NIRS signal, with the maximum value representing 100% tissue oxygenation. The short-duration NMES was used to elevate mVO2 without inducing an oxygen debt to minimize any discomfort to the participants. A stimulation check was performed for all tests. This calibration was used to scale NIRS signals to this maximal “physiological” range, thus facilitating accurate comparisons between individuals with varying adipose tissue thickness (ATT).

Biometric data, including age, skin color, hair color, hair thickness, height, weight, and body mass index (BMI), were collected. Height and weight were used to calculate BMI. The left leg was used for all participants. ATT of the left biceps femoris, vastus lateralis, medial gastrocnemius, and tibialis anterior were measured at the end of each NIRS protocol at the site of the NIRS optode in the fowler’s position with ultrasound (Butterfly IQ+, Butterfly Network Inc, Burlington, MA, USA).

### 2.3. Analysis

mVO_2_ was measured as the rate of change in SmO_2_ signals during brief arterial occlusions. Changes in mVO_2_ with stimulation were evaluated by comparing the post-stimulation and recovery conditions with the pre-stimulation conditions. The means and standard deviations for slopes and metabolic rates were calculated for each muscle. Paired *t*-tests were used for the between-group comparison of the metabolic rates of the various muscle groups. Significance was accepted when *p* < 0.05.

## 3. Results

The physical characteristics of participants are shown (Table 1). All subjects (*n* = 20) were able to complete testing with no adverse events. Subjects ranged between skin color types 1–3 on the Fitzpatrick scale. The stimulation intensity was maintained throughout the procedure in all subjects. Four measurements failed out of the eighty muscle groups stimulated (four muscles * twenty subjects); three on the VL and one on the BF muscle. The ATT values in those muscles were 18.7, 13.0, 13.2, and 10.6 mm, respectively. There were significant differences in average ATT over the muscle tested between males and females (*p* = 0.03). The mVO_2_ results for the men and women were combined, as there were no sex differences in this measure for the combined muscle groups and only a difference in one individual muscle group (MG, *p* = 0.003). See Appendix A for more details on sex comparisons.

Representative data for the VL muscle is presented (Figure 2). The average mVO_2_ values for the four muscle groups are shown (Figure 3), both as absolute values and as ratios of resting muscle metabolism. mVO_2_ was significantly elevated at both the 5 and 10 min time points (*p* < 0.001 for both). Five minutes post-stimulation mVO_2_ was not different from the initial resting values (*p* = 0.31). The average mVO_2_ for each muscle tested is presented (Table 2).

A histogram of individual responses for the average mVO_2_ is shown (Figure 4). Nine subjects achieved the target value of 8-fold or greater increases in mVO_2_ for the average of the five- and ten-minute time points. There were no differences between the high and low mVO_2_ groups in ATT (8.2 to 8.9, *p* = 0.77). There were small differences in the number of men or women between the high-responding group (six women, three men) and the low-responding group (eight women, three men). There was no difference in BMI between high and low responding groups (*p* = 0.50).

## 4. Discussion

The primary finding of this study was that electrical stimulation of four muscles produced, on average, a 6.6-fold increase in muscle metabolism from resting metabolic rate. NMES has been previously applied to various muscle groups, including the medial gastrocnemius, quadriceps femoris, vastus lateralis, shoulder flexors, elbow extensors, wrist and finger extensors, thumb abductors, biceps, and forearm muscles [26,27,28]. Studies using multiple muscle group stimulation typically do not report an increase in specific muscle metabolism. The current study used a stimulation rate of 3 Hz. Stimulation rates of 3 Hz have been used previously [29]. Ryan et al. [30] found a 13-fold increase in muscle metabolism with 3 Hz stimulation, which is greater in metabolism than what was found in this study. Ryan et al. [30] also used higher stimulation frequencies, up to 6 Hz, which produced even higher metabolic rates. The higher metabolic rates compared to our study could be due to using higher stimulation currents. Other studies of one muscle group have reported that an 8–10-fold increase in muscle metabolism was usually obtained, again with 30 s of stimulation of one muscle group [31]. Subjects in those studies needed to tolerate 30 s of stimulation of one muscle group, compared to 10 min of stimulation of four muscle groups in our study. Our study used ten minutes of electrical stimulation to evaluate steady levels of muscle metabolism; however, the eventual goal of multiple muscle stimulation would be to match guidelines for exercise training of 30 min or more of aerobic exercise [32]. The increase in metabolic rate in this study was comparable to the metabolic rate found in the VL with walking at 3.2, 4.8, and 6.4 km/h [33]. This suggests that the changes in local muscle metabolism in this study were comparable to mild to moderate voluntary exercise. One of the limitations of the current study was that it did not collect measurements of whole-body changes in metabolism, such as heart rate, whole-body oxygen consumption, blood glucose levels, or cardiac output.

The magnitude of the increase in mVO_2_ depends on both the stimulation rate and the stimulation current. We chose twitch contractions because they result in lower force levels compared to higher stimulation rates because of the lack of summation and tetanus. Previous research predominantly focuses on the use of tetanic muscle stimulation, which poses challenges for metabolic rate assessment and tends to yield outcomes related to pain relief or limb movement [34]. The twitch contractions include a large concentric component as the duration of contraction is not long enough to allow for limb movement or the development of isometric force. Concentric contractions use more energy than isometric contractions, allowing for higher energy use relative to force development [35]. Twitch contractions can also produce less pain sensations and be better tolerated by the subjects. The lower mVO_2_ values in this study compared to single-muscle studies may result from using lower current levels to stimulate four muscle groups instead of one. Direct comparison of current levels is challenging in part due to differences in electrode types (aluminum foil versus commercial electrodes), sizes, placements, and skin preparations across studies. We selected current levels that subjects could tolerate for 10 min.

A key aspect of this study was the use of economical stimulation equipment. We developed the experimental setup by exploring various configurations and adjusting different stimulation settings. The protocol utilized an affordable, commercially available four-output, two-channel stimulation device priced at approximately $80 USD, along with aluminum foil electrodes. The total equipment cost was below $100 USD for a single leg and under $200 USD if used for both legs. The low cost of the equipment significantly reduces barriers to entry for both research and clinical applications. This cost-efficiency is particularly advantageous for large-scale studies, enabling the inclusion of diverse cohorts without excessive financial burden. In clinical practice, affordability could make routine NMES interventions more feasible in outpatient settings, rehabilitation centers, and even home-based therapy programs, where cost considerations often limit treatment options. The primary advantage of this technique is its ease of use and the portability of the associated technology. Additionally, the simplicity of the device reduces the need for specialized training, allowing healthcare providers and even patients or caregivers to administer NMES with minimal supervision. Previous studies on one muscle group used high-grade electrical stimulators, which can cost $1700 USD or more, in addition to using adhesive commercial electrodes that cost $3–5 per pair of electrodes. The use of an inexpensive electrical stimulation device does come with some disadvantages. The current levels are not as easy to adjust and often are presented as ‘clicks’ or numbers from 1 to 10 (low to high), rather than in units of mAmps. The use of aluminum electrodes makes the cost of the electrodes much lower, although they did require connectors from old self-adhesive electrodes. These reusable connectors significantly reduced costs. Water-based reusable electrodes could also be used to reduce expenses.

Approximately half (9 of 20) of the subjects had an 8-fold increase in mVO_2_ above the resting target. Some subjects had very low metabolic rates, with 2–4-fold increases. This was due to tolerating lower stimulation intensities. Some previous studies have implemented familiarization protocols for electrical stimulation and related measurements to reduce misunderstandings, particularly concerning muscle contractions induced by electrical stimulation [36,37,38]. Although a stimulation check was conducted, the protocol used in this experiment did not include a familiarization phase. We hypothesize that incorporating a familiarization session could have allowed participants to tolerate higher current intensities. This hypothesis is supported by the data, which show that participants with prior experience in electrical stimulation (approximately four individuals) were able to tolerate higher currents compared to those without such experience. Subjects in this study often found electrical stimulation unfavorable due to its unfamiliarity. As a result, the electrical stimulation was administered at a reduced maximum tolerable intensity. However, this issue may be less relevant for the patient population due to reduced sensitivity in their lower limbs.

Some people who might benefit from multiple muscle stimulation have reduced or no sensory function in their muscles. NMES has been extensively studied in patients with motor and sensory complete spinal cord injuries, including those with paralyzed abdominal muscles. Research indicates that patients report higher satisfaction levels following NMES treatment compared to a placebo-controlled group [39]. Furthermore, evidence suggests that participants generally prefer a stimulation frequency of 3 Hz over 1 Hz, potentially due to the enhanced ability to fully potentiate the muscle [29]. Despite these findings, there remains a notable lack of discussion on participants’ subjective responses to NMES, particularly regarding discomfort or pain. This is especially relevant as athletes are often reluctant to use NMES due to the associated discomfort or pain [40]. This could deter consistent use, undermining the therapeutic benefits of NMES. Therefore, developing and testing devices that minimize discomfort is crucial for improving compliance and ensuring the effectiveness of treatment regimens in target populations. Nevertheless, numerous studies have demonstrated that NMES can enhance athletic performance, improve muscle capacity, and aid recovery by reducing perceived soreness and lowering creatine kinase concentrations [41].

The benefits of exercise extend to several different physiological changes within the body. These include changes in muscle metabolism, as well as arterial blood flow, energy metabolism, and cardiac work [42]. Although low force twitch contractions are not expected to induce muscle hypertrophy or ward off sarcopenia, they can provide metabolic benefits and improve muscle endurance. These studies have reported changes in whole body metabolism [37].

There are several limitations to consider when evaluating the conclusions of this study. Firstly, the protocol involved testing only one leg, which raises the question of whether including a second leg would influence the results. It is hypothesized that testing both legs could increase the intensity of the stimulation and the duration of the procedure, especially if electrodes are applied manually. Future studies should aim to develop faster and more efficient application methods. However, based on our findings, we extrapolate that testing a single leg provides sufficient data. Another limitation is that the study was conducted on a small cohort of healthy individuals, which limits the generalizability of our findings to populations with varying muscle conditions, such as muscle atrophy or aging-related sarcopenia. Despite this, our sample demonstrates skeletal muscle metabolism responses under electrical stimulation, supporting the feasibility of using this method to induce metabolic changes in targeted muscle groups. Future studies should expand the sample to include a broader age range and individuals with muscular pathologies to better evaluate the method’s applicability and impact in diverse populations. Additionally, the potential influence of ATT on the results could affect the amount of light detected by the NIRS device. Despite this, we obtained adequate NIRS signals, suggesting that ATT may not have significantly impacted our findings. The device used in this study was limited to two channels, each capable of stimulating two areas at the same intensity and frequency. This limitation may have resulted in lower tolerable intensity in muscle groups that could otherwise handle higher levels of stimulation. However, since most participants tolerated a consistent intensity across all four muscle groups, the restriction on the number of available channels does not appear to have significantly affected the results. Furthermore, future studies could explore modifications or complementary tools to optimize the device’s performance.

Moreover, no obese participants (BMI > 30) were included in the study, and only four participants with a BMI indicative of being overweight (25.0 < BMI < 29.9) were tested. Of these participants, 75% exhibited relatively lower activation of the muscle groups. To our knowledge, no other studies have specifically addressed the impact of ATT or BMI on NMES results. Generally, ATT and BMI do not show a significant difference in outcomes, especially in our relatively young and thin sample population. However, given that many participants were not acclimatized to electrical stimulation, similar to the general population, it is possible that an older population with higher ATT might experience greater difficulty or discomfort. Future protocols could incorporate familiarization sessions to address this, and larger electrodes could be developed to distribute the current over a broader area.

Future studies should focus on linking increased metabolic rates and electrical stimulation to health benefits. This could determine the target levels of muscle metabolism needed in therapeutic programs. Future research is required to design an easier approach to applying electrodes, especially for people with disabilities. Because some people consider electrical stimulation uncomfortable at almost all current levels, approaches to increasing comfort levels or tolerance of the electrical stimulation will also be needed.

## 5. Conclusions

In conclusion, the findings of this study demonstrate that a cost-effective approach to NMES can be a potentially valuable method for enhancing muscle metabolic activity, as measured by NIRS. The observed 6-fold increases in muscle metabolism have the potential to provide health benefits if performed as a training regimen. The ease of use and low cost of this device make it an excellent candidate for larger cohort studies, including those targeting clinical populations with muscle atrophy or metabolic conditions. Additionally, its economic feasibility supports longitudinal studies, enabling the long-term monitoring of metabolic and functional improvements in patients. Future research should focus on the application of this technique within clinical populations, with an emphasis on the development of accessible therapeutic devices and equipment to facilitate widespread clinical use.

## Figures and Tables

**Figure 1 jfmk-09-00248-f001:**
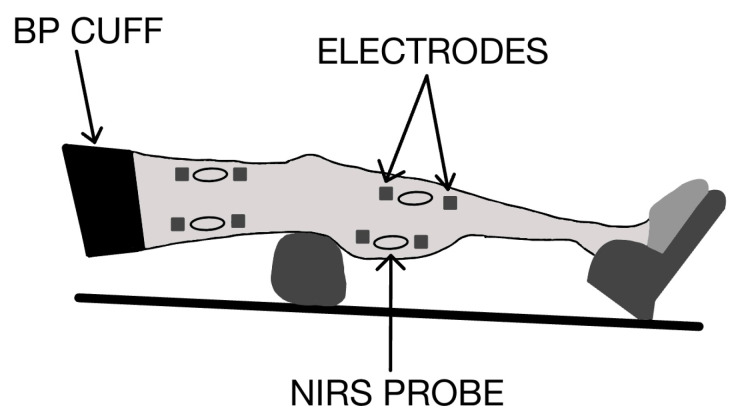
The experimental setup for measurements on the four muscles using near-infrared spectroscopy (NIRS). Aluminum foil electrodes were placed proximally and distally. The black sleeve is an inflatable blood pressure cuff, placed on the upper thigh, for rapid cuff inflation. The participant’s foot rests on the pedal and receives support from a thin pillow placed below the knee.

**Figure 2 jfmk-09-00248-f002:**
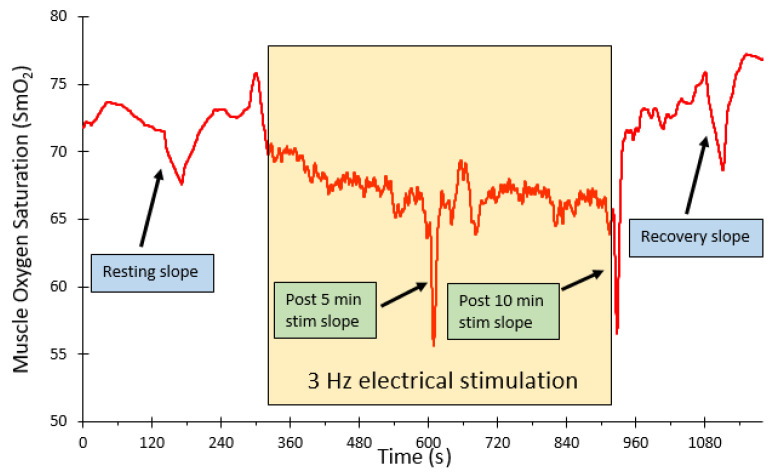
Representative example of the vastus lateralis muscle oxygen saturation during rest, resting arterial occlusions, 5 min neuromuscular electrical stimulation exercises, and end-exercise recovery. A 5 s arterial occlusion is performed after each period of electrical stimulation to determine the relative oxygen levels.

**Figure 3 jfmk-09-00248-f003:**
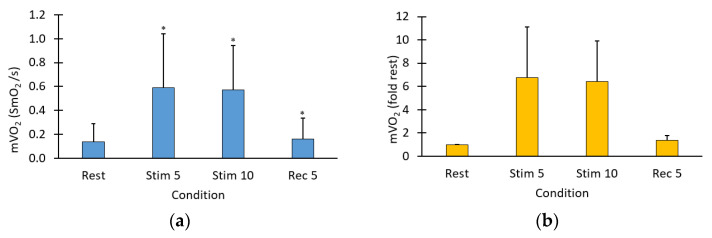
(**a**) Average muscle oxygen saturation for all four muscles throughout the procedure. Values are means (STDEV). Statistical differences were found for all comparisons to rest in this figure; (**b**) average metabolic rates for all four muscles during the different conditions. The normalized value is calculated as a ratio of the resting slope and the stimulation slope. Values are means (STDEV). * *p* < 0.05.

**Figure 4 jfmk-09-00248-f004:**
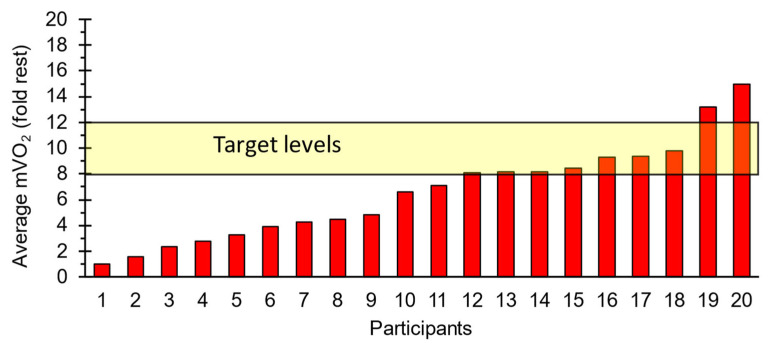
A histogram of average increases in mVO2 for 5 and 10 min for the four muscle groups. The target goal of an 8–12-fold increase is shown by the yellow shaded area.

**Table 1 jfmk-09-00248-t001:** Physical characteristics of participants.

Age, y	BMI, kg/m^2^	Sex, M/F	VL ATT, mm	BF ATT, mm	TA ATT, mm	MG ATT, mm	Average ATT, mm
20.4	23.3	6 M/14 F	9.7	8.5	6.7	6.8	8.5
(0.7)	(3.4)		(3.8)	(3.5)	(1.6)	(2.3)	(2.7)
20.0	22.7	Female	10.8	9.0	7.3	7.9	9.4
(1.1)	(2.8)		(3.6)	(3.3)	(1.5)	(1.8)	(2.4)
*					*	*	*
21.3	24.7	Male	7.4	7.5	5.3	4.6	6.6
(0.5)	(4.5)		(3.2)	(3.9)	(0.9)	(2.0)	(2.2)

Values are expressed as means (SD). M, male; F, female; ATT, adipose tissue thickness; VL, vastus lateralis; BF, biceps femoris; TA, tibialis anterior; MG, medial gastrocnemius. * *p* < 0.05 between males and females.

**Table 2 jfmk-09-00248-t002:** mVO_2_ values relative to resting levels for the different muscle groups. Values are means (SD).

	Increases in Muscle Metabolism			
Muscle	Rest (Ratio)	Stim 5 min	Stim 10 min	Rec 5 min
Vastus Lateralis	1	8.9 (8.6)	9.4 (7.4)	1.9 (0.9)
Medial Gastrocnemius	1	6.6 (7.8)	7.1 (7.9)	1.5 (0.6)
Biceps Femoris	1	7.9 (11.9)	5.6 (5.8)	1.2 (0.9)
Tibialis Anterior	1	4.9 (3.9)	4.9 (3.8)	1.2 (0.5)

## Data Availability

Data are available on request.

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
