# Peer review of "Muscle Metabolism During Multiple Muscle Stimulation Using an Affordable Equipment"

_jfmk, 2024, doi:10.3390/jfmk9040248_

Round 1
Reviewer 1 Report
Comments and Suggestions for Authors
A comprehensive work has been presented by the authors, with the objective of quantifying the increases in muscle metabolism in four muscles of the legs of able-bodied individuals with NMES. It is a work that is pertinent to the practical application of health sciences. In the following section, I identify potential enhancements.
Abstract: The abstract is extremely precise and explicit, as it employs headings prior to the objective, methodology, and conclusions sections, which facilitates comprehension.
Introduction: The work of NMES and NIRS could be more effectively presented to provide a more comprehensive explanation of the systems and their functions.
Materials: The male/female percentage and other details, such as the reasons for the others' exclusion and the definition of "healthy," can be enhanced in the participant description.
Results: The figures and tables facilitate comprehension of the data, and I regard them as well-written.
Discussion: I am of the opinion that she could provide a separate perspective on practical applications and future directions.
Conclusion: I would require additional development.
References: There are references from over two decades ago that I am uncertain about their potential for updating. However, I acknowledge that they may be able to be updated, as they are currently in use.
Author Response
A comprehensive work has been presented by the authors, with the objective of quantifying the increases in muscle metabolism in four muscles of the legs of able-bodied individuals with NMES. It is a work that is pertinent to the practical application of health sciences. In the following section, I identify potential enhancements.
We appreciate the time and effort the reviewer has spent to evaluate our paper.
Abstract: The abstract is extremely precise and explicit, as it employs headings prior to the objective, methodology, and conclusions sections, which facilitates comprehension.
Thank you.
Introduction: The work of NMES and NIRS could be more effectively presented to provide a more comprehensive explanation of the systems and their functions.
We have included an additional reference on the NIRS method to help the reader understand NIRS and its potential to study skeletal muscle.
Materials: The male/female percentage and other details, such as the reasons for the others' exclusion and the definition of "healthy," can be enhanced in the participant description.
We have revised the paper to include a better description of ‘healthy’. The number of male and female subjects is included in table 1.
Results: The figures and tables facilitate comprehension of the data, and I regard them as well-written.
Thank you.
Discussion: I am of the opinion that she could provide a separate perspective on practical applications and future directions.
The revised paper includes additional discussion of the value of low cost stimulation equipment and potential health benefits.
Conclusion: I would require additional development.
The discussion and conclusion sections have been revised to better highlight the significance of the study as well as several potential limitations.
References: There are references from over two decades ago that I am uncertain about their potential for updating. However, I acknowledge that they may be able to be updated, as they are currently in use.
The references have been revised to include several more recent papers as well as to reduce the number of self citations.
Reviewer 2 Report
Comments and Suggestions for Authors
It is an interesting and impactful study for people with various pathologies, who cannot move or do physical exercise, due to muscle atrophy. However, such studies have been done before, both on healthy people and on patients. The situation in itself is not new and the disadvantage of this study is that a larger and more varied cohort of people was not used, such as a larger age range, with an affected muscular system that ages and supplemented even with patients with atrophy muscular as the first symptom or as a secondary symptom of another condition.
This is rather a pilot study to test the method of electrical stimulation on 4 muscles to monitor metabolic changes in healthy people, but with different degrees of muscle training. It is important that the method avoids pain, but improves the metabolism of the skeletal muscle, using an economical stimulation equipment.
It would be desirable to evaluate the equipment used on a larger number of people, both healthy and those with different pathologies who would benefit from muscle stimulation. Thus, the scientific conclusions will be closer to the truth and more valuable.
Author Response
It is an interesting and impactful study for people with various pathologies, who cannot move or do physical exercise, due to muscle atrophy. However, such studies have been done before, both on healthy people and on patients. The situation in itself is not new and the disadvantage of this study is that a larger and more varied cohort of people was not used, such as a larger age range, with an affected muscular system that ages and supplemented even with patients with atrophymuscular as the first symptom or as a secondary symptom of another condition.
This is rather a pilot study to test the method of electrical stimulation on 4 muscles to monitor metabolic changes in healthy people, but with different degrees of muscle training. It is important that the method avoids pain, but improves the metabolism of the skeletal muscle, using an economical stimulation equipment.
It would be desirable to evaluate the equipment used on a larger number of people, both healthy and those with different pathologies who would benefit from muscle stimulation. Thus, the scientific conclusions will be closer to the truth and more valuable.
We appreciate the time and effort the reviewer has taken to evaluate our paper. We agree with the reviewer that previous studies have evaluated the health benefits of electrical stimulation. In addition, stimulation of multiple muscle groups has also been performed. However, we feel our paper has two significant differences from previous papers. First is the measurement of muscle metabolic rate during multiple muscle stimulation. Previous studies have focused on whole body responses to stimulation, or have measured muscle metabolism in response to a very specific and short duration stimulation. The presentation of muscle metabolic rates with multiple muscle stimulation we feel is unique.
In addition, our study focused on the use of inexpensive equipment and supplies. Prior studies have not focused on the cost of their equipment/supplies. Most studies the authors are aware of used stimulation devices that cost between $1,500-$6,000 US dollars. These studies also used self-adhesive stimulation electrodes that cost $4-6 US dollars per pair of electrodes. Finally, many of the multiple muscle stimulation papers used expensive control units that can cost more than $20,000 US dollars. The use of equipment that costs less that $200 US dollars is a significant lower cost and should encourage greater utilization of the method.
We agree that further studies should evaluate the use of this equipment on more subjects to further test its potential. This has been included in the limitations and future direction sections of the discussion.
Round 2
Reviewer 2 Report
Comments and Suggestions for Authors The authors have added some clarifications and specifications that improve the presentation. Although I maintain that the scientific data are not very valuable, the presentation of the new neuromuscular electrical stimulation (NMES) procedure deserves the attention of readers and especially of medical professionals in the targeted field. It is important that the authors confirm that they will continue studying this method, enabling a long-term monitoring of metabolic and functional improvements both on a larger number of healthy subjects, but especially on patients.